# Immune Thrombosis: Exploring the Significance of Immune Complexes and NETosis

**DOI:** 10.3390/biology12101332

**Published:** 2023-10-12

**Authors:** José Perdomo, Halina H. L. Leung

**Affiliations:** 1Haematology Research Group, Faculty Medicine and Health, Central Clinical School, University of Sydney, Sydney, NSW 2006, Australia; 2Haematology Research Unit, St George & Sutherland Clinical Campuses, Faculty of Medicine & Health, School of Clinical Medicine, University of New South Wales, Kogarah, NSW 2217, Australia; halina.leung@unsw.edu.au

**Keywords:** immune thrombosis, immune complex, neutrophil extracellular traps

## Abstract

**Simple Summary:**

While neutrophil extracellular traps (NETs) are released by neutrophils as part of immune processes, there is extensive evidence of the involvement of NETs in pathological developments such as thrombosis. Immune complexes, formed by antibodies against diverse antigens, can trigger NETs formation, which induces a generation of neutrophil–platelet aggregates, platelet activation, vascular damage, and thrombus formation. Thromboses associated with autoimmune diseases and severe infections are thought to be caused by immune complex-induced NETs formation. Here, we explore the impact of immune complexes on NETs-associated thrombosis.

**Abstract:**

Neutrophil extracellular traps (NETs) are major contributors to inflammation and autoimmunity, playing a key role in the development of thrombotic disorders. NETs, composed of DNA, histones, and numerous other proteins serve as scaffolds for thrombus formation and promote platelet activation, coagulation, and endothelial dysfunction. Accumulating evidence indicates that NETs mediate thrombosis in autoimmune diseases, viral and bacterial infections, cancer, and cardiovascular disease. This article reviews the role and mechanisms of immune complexes in NETs formation and their contribution to the generation of a prothrombotic state. Immune complexes are formed by interactions between antigens and antibodies and can induce NETosis by the direct activation of neutrophils via Fc receptors, via platelet activation, and through endothelial inflammation. We discuss the mechanisms by which NETs induced by immune complexes contribute to immune thrombotic processes and consider the potential development of therapeutic strategies. Targeting immune complexes and NETosis hold promise for mitigating thrombotic events and reducing the burden of immune thrombosis.

## 1. Introduction

Morphological changes in neutrophils, distinct from apoptosis or necrosis, upon treatment with phorbol 12-myristate 13-acetate (PMA) were described by Takei et al. in 1996 [1]. This process was later termed neutrophil extracellular traps (NETs) formation and was initially reported as a host defence mechanism against infection [2]. NETs formation has evolved into a mechanism with wide implications in biological processes and in the pathology of diabetes, cancer, autoimmunity, atherosclerosis, infections, and thrombosis. Two distinct types of NETosis have been described: a slow form of cell death that may take hours and is known as lytic or suicidal NETosis and vital NETosis, a fast process whereby neutrophils release nuclear material but remain viable and functional [3]. In fact, enucleated neutrophils are known to retain chemotactic and phagocytic functions [4]. 

Lytic NETosis depends on NADPH and reactive oxygen species (ROS) activity which triggers histone citrullination via peptidyl arginine deiminase (PAD4), while vital NETosis does not involve NADPH and relies on increases in intracellular calcium [5]. A sub-category of lytic NETosis, termed noncanonical NETosis, has been proposed more recently [6]. This form of NETosis is elicited in the presence of cytosolic Gram-negative bacteria and involves noncanonical inflammasomes and caspase 4/5 activation. Histone citrullination mediated by PAD4 is also present in noncanonical NETosis [7]. Both lytic and vital NETosis, however, involve nuclear the decondensation and subsequent release of chromatin into the extracellular space. Chromatin serves as the backbone for numerous NET components including histones, myeloperoxidase, elastase, and cathepsin-G [8]. Therefore, the consequences of NETs formation—antibacterial activity, inflammation, vascular occlusion—can be attributed to the activity of the diverse components of these DNA structures.

Since the description of NETs induction by PMA and bacteria [1,2], numerous stimuli of NETs formation such as viruses [9], cholesterol crystals [10], activated platelets [11,12], cytokines [13], autoantibodies [14], and immune complexes (ICs) [15,16,17] have been described. Unsurprisingly, an attendant number of neutrophil receptors have been associated with various stimuli. Activated platelets induce NETosis via the PSGL1 receptor [15,18] or via high-mobility group protein B1-RAGE interaction [19], bacteria and viruses through pattern recognition receptors and Fc receptors [20,21] and ICs via Fc receptors [15,16,22] (Figure 1).

## 2. Immune Complexes and Fc Receptors

The adaptive immune response results in the production of antibodies against antigens, leading to the formation of antigen–antibody complexes. The generation of antibodies that bind to different epitopes on the antigen leads to the formation of ICs. The role of ICs in conditions such a serum sickness, vasculitis, and rheumatic disease has been documented for decades [23]. ICs have a higher affinity for Fc receptors and are normally cleared by the liver (by sinusoidal endothelial cells and Kupffer cells) and spleen via interaction with Fc receptors expressed on monocytes/macrophages and neutrophils. Neutrophils express both low- (FcγRIIA, FcγRIIB, and FcγRIIIB) and high- affinity (FcRn and FcγRI, which is expressed on activated neutrophils) IgG Fc receptors. FcαRI, a receptor for IgA, is also present on human neutrophils [24]. High-affinity receptors such as FcγRI bind IgG monomers, while the low affinity receptors have an avidity for IgG ICs or opsonised cells. IgG subclasses display distinct Fc receptor binding affinities. IgG1 and IgG3 interact with FcγRI, FcγRIIA/B, FcγRIIIB, and FcRn, while IgG2 fails to recognise FcγRI, FcγRIIB, and FcγRIIIB [25]. Some IgG2 and IgG4 ICs also recognise complement receptors on neutrophils [26]. 

The recognition of ICs by Fc receptors leads to cellular signalling. For example, the clustering of FcγRIIA via interaction with ICs induces a phosphorylation of the immunoreceptor tyrosine-based activation motif (ITAM) by Src kinases [27], leading to Syk signalling involving PI3K and PLCγ. This triggers diverse responses including phagocytosis and receptor internalisation, cytokine production, and oxidative burst. Murine neutrophils expressing human FcγRIIA and FcγRIIIB are able to uptake ICs, but only FcγRIIA and not FcγRIIIB transgenic mice formed NETs in response to ICs in vivo [28]. In human neutrophils, however, FcγRIIIB has been found to be involved in NETs formation [29]. 

Circulating ICs resulting from excessive antibody production or clearance failure can be deposited on tissues, causing inflammation, and as such are involved in the pathology of multiple conditions including autoimmune diseases such as arthritis and lupus erythematosus. Drug-mediated reactions such as heparin-induced thrombocytopenia (HIT) and vaccine-induced thrombotic thrombocytopenia (VITT) and infections including influenza and SARS-CoV-2 are facilitated by circulating ICs. In fact, the induction of platelet release via IC activity has been known since the 1950s [30].

Since the topic of this review is the contribution of NETs to thrombotic processes, the role of ICs on other Fc-expressing granulocytes is not discussed in detail. However, it should be noted that ICs also induce processes analogous to NETosis in monocytes [31]. Monocyte extracellular traps (ETs) possess a procoagulant activity, suggesting a role for these monocyte-derived structures in thrombosis (reviewed by Han et al. [32]). The more generalised nature of ET formation and its association with thrombosis is exemplified by observations of ETs of macrophage, mast cell, and eosinophil origin in coronary thrombi [33]. 

IgG, and more specifically IgG1, is the most common type of immunoglobulin in human serum [25]. IgG ICs have been more widely described, and it is reasonable to assume that IgG ICs are also the most abundant. Lupus erythematosus is an extensively studied autoimmune disease characterised by the presence of autoantibodies against numerous endogenous antigens including dsDNA, ribonucleoprotein, phospholipids, histones, and β2-glycoprotein I. Most of these autoantibodies are of the IgG class [34]. DNA-containing Ics found in lupus patients signal through FcγRIIA and induce ROS production in neutrophils [35]. This is reminiscent of ROS-dependent neutrophil activation, NETosis, and thrombosis induced by FcγRIIA-activating HIT Ics [36]. 

ADAMTS-13 Ics in thrombotic thrombocytopenic purpura are correlated with relapse [37] and markers of NETosis such as histone/DNA complexes, cell-free DNA, and citrullinated histones are present in plasma from these patients [38], suggesting a contribution of ICs in the inductions of NETosis and thrombosis in this condition. IgG ICs are also present in granulomatosis (antineutrophil cytoplasmic antibodies [39]), SARS-CoV-2 infection [40], rheumatoid arthritis [41], HIT, and VITT [15,16] (Table 1). 

Thrombosis is also driven by IgA ICs (Table 1). Antibody specificity and signalling is dependent on the isotype, where IgG binds to FcγR while IgA binds to FcαR. In the context of IgG signalling, unlike low-affinity FcγRs, only high-affinity FcγRs (i.e., FcγR1) can bind monomeric IgG. All FcγRs can, however, bind to IgG aggregates or immune complexes containing IgG [42]. Similarly, monomeric IgA binds poorly to FcαR1, while large IgA complexes bind with a high avidity, leading to phagocytosis, antigen presentation [43], cytokine release [44], reactive oxygen species production [45], and NETosis [46,47,48]. 

The proinflammatory IgA immune complex-mediated FcαR signalling is a key pathogenic feature in IgA vasculitis (or Henoch–Schönlein purpura), an inflammatory condition where the immune system attacks the lining of blood vessels. Recently, Mayer-Hain et al. showed the requirement of neutrophil prestimulation by polymeric IgA or IgA ICs to lower the threshold for neutrophil activation. This step is critical for neutrophils to become activated and undergo NETosis upon binding to activated endothelial cells, resulting in vessel wall damage [46]. Interestingly, neutrophils isolated from IgA vasculitis patients spontaneously underwent NETosis, and NETs were proximal to endothelial cells and IgA-coated neutrophils in tissue sections of these patients [46]. A significant reduction in vessel damage in a mouse model of vasculitis was observed following NET inhibition, suggesting that NETosis is key a mediator of vasculitis pathogenesis. Although uncommon, cases of coagulation abnormalities and thrombosis have been documented in IgA vasculitis [49]. Altogether, IgA ICs prestimulate neutrophils and NETosis is a key mediator of vessel wall damage in IgA vasculitis.

The presence of IgA autoantibodies is also associated with increased disease severity, enhanced cartilage damage, and worse disease prognosis in rheumatoid arthritis [50,51,52]. The activation of neutrophils by IgA ICs, present in rheumatoid arthritis patients’ plasma and synovial fluid, leads them to undergo NETosis and secrete chemoattractants that amplify neutrophil recruitment [47] and promotes cartilage damage via neutrophil elastase [48]. IgA IC-induced NETosis can be blocked by anti-FcαR1 monoclonal antibody [47], suggesting FcαR1 inhibitors could potentially reduce cartilage damage and disability in rheumatoid arthritis patients. In antiphospholipid syndrome (APS), β2-glycoprotein I/IgA IC is strongly linked to thrombosis following transplantation [53]. NETosis is likely to be of pathological significance since NETs are known to contribute to thrombosis in APS [54].

NETosis has been well documented in bacterial (e.g., *Staphylococcus aureus* [55]) and viral (e.g., influenza A, HIV, SARS-CoV-2 [21]) infections. NETs exert both antimicrobial [56] and antiviral [21] activity. IgA-bacteria ICs activate FcαRI and enhance the phagocytosis of IgA-opsonized bacteria [57]. Interestingly, the phagocytosis of *S. aureus* is more efficient in the presence of IgA [58]. Both bacterial and viral IgA IC-induced NETosis are dependent on FcαRI and NADPH oxidase [21,58]. A lower virus titre is required to trigger NETosis in the presence of IgA compared to virus alone, and NETosis is independent of phagocytosis. Unlike NETosis induced by virus alone, IgA-virus ICs-induced NETosis does not require toll-like receptor signalling [21]. Although NETosis plays a protective role in infection, the release of reactive oxygen species, proteolytic enzymes and inflammatory mediators can enhance neutrophil infiltration and elicit tissue damage. This can result in the enhancement of disease pathogenesis in conditions such as respiratory syncytial virus [59], rhinovirus [60], influenza [61], and COVID-19 [62,63,64].

**Table 1 biology-12-01332-t001:** Fc receptors implicated in IgG- and IgA-immune complex-mediated NETosis.

Condition	Fc Receptor Involved	Receptor Binding Specificity	Antigen-Antibody Complex	Reference
Influenza A	FcαR1 (CD89)	IgA	Influenza—IgA	Stacey et al. [21]
Human immunodeficiency virus (HIV)	FcαR1 (CD89)	IgA	HIV—IgA	Stacey et al. [21]
SARS-CoV-2	FcαR1 (CD89) FcγRIIA (CD32a)	IgA IgG	SARS-CoV2—IgA Spike protein—IgG	Stacey et al. [21] Bye et al. [65]
Staphylococcus aureus	FcαR1 (CD89)	IgA	*S. aureus*—IgA	Aleyd et al. [58]
Vasculitis	FcαR1 (CD89)	IgA	Aggregated vasculitis—IgA	Mayer-Hain et al. [46]
Rheumatoid arthritis	FcαR1 (CD89), FcγRI (CD64), FcγRIIA (CD32a)	IgA, IgG	Rheumatoid factor—IgA/IgG, citrullinated protein—anticitrullinated protein antibody (ACPA) IgG, cyclic citrullinated peptide—IgA/IgG, antineutrophil cytoplasmic antibodies (ANCA)—IgA/IgG	Mathsson et al. [41]; Aleyd et al. [47]; Kempers et al. [66]
Granulomatosis with polyangiitis (Wegener’s granulomatosis)	FcαR1 (CD89), FcγRIIIB (CD16b)	IgA, IgG	Antineutrophil cytoplasmic antibodies (ANCA)—IgA/IgG	Kelley et al. [39]
Heparin-induced thrombocytopenia (HIT)	FcγRIIA (CD32a)	IgG	Heparin—PF4 -HIT IgG	Kelton et al. [67]; Chong et al. [68]
Vaccine-induced thrombotic thrombocytopenia (VITT)	FcγRIIA (CD32a)	IgG	PF4—VITT IgG	Greinacher et al. [69]
Autoimmune inflammatory disorder	FcγRIIA (CD32a), FcγRIIIB (CD16b)	IgG	Bovine serum albumin (BSA)—IgG, human serum albumin (HAS)—IgG, cross linking—FcγRIIIB	Aleman et al. [29]; Behnen et al. [22]
Systemic lupus erythematosus (SLE)	FcγRIIA (CD32a)	IgG	DNA—IgG	Bonegio et al. [35]; Bruneau et al. [70]; Patiño-Trives et al. [71]; Dema and Charles [34].

## 3. Mechanisms of Neutrophil Activation

The expression of Fc receptors by neutrophils is indicative of the role of these receptors in NETs formation. Chen and colleagues first showed that the engagement of FcγRIIA by soluble ICs resulted in NETosis in transgenic mice [28]. In this setting, FcγRIIIb acted in the removal of ICs by internalization but did not lead to neutrophil activation. Nevertheless, Alemán et al. showed that FcγRIIIB cross-linking with a monoclonal antibody induced NETs formation in human neutrophils [29]. FcγRIIIB-mediated activation involved ERK1 and transforming growth factor-β-activated kinase 1 phosphorylation [29]. The same authors demonstrated that neither cross-linking of FcγRIIA with the IV.3 monoclonal antibody nor activation with PMA induced transforming growth factor-β-activated kinase 1 phosphorylation, despite the induction of ERK1 phosphorylation by PMA [29]. This illustrates the various signalling pathways that lead to NETs formation following neutrophil activation. Immobilised ICs also induced NETs via FcγRIIIB in a NADPH-dependent manner involving the Src/Syk pathway [22], suggesting a role in NETs formation for both FcγRIIA and FcγRIIIB (Figure 1). These observations did not define whether both receptors are required or if there is a main receptor implicated in NETosis. In the case of ICs from HIT patients, however, Xiao et al. showed neutrophil activation via FcγRIIA without the involvement of FcγRIIIB [72]. Consistent with these findings, IgG ICs from HIT and VITT patients were shown to engage FcγRIIA exclusively to promote NETs formation and thrombosis [15,16]. HIT IC-FcγRIIA interaction led to ROS production in both platelets and neutrophils. The potential clinical relevance of these findings was supported by observations of increased ROS production in activated neutrophils (known as low-density granulocytes) from patients with active HIT. Moreover, ROS inhibition abrogated NETs formation in human blood and was sufficient to block thrombosis in a mouse model of HIT [36].

Platelets have long been considered significant contributors of thrombosis. This led to the development and wide use of various antiplatelet drugs as primary and secondary prevention of recurring thrombotic complications, particularly following myocardial infarction and ischemic stroke. Platelets can be activated by various stimuli such as immune complexes [16], viruses (e.g., SARS-CoV-2) [73], and vascular injury [74] resulting in either a direct or indirect activation of neutrophils. With the former, activated platelets can bind and activate neutrophils via various corresponding receptors such as P-selectin/PSGL1, GP1bα/Mac-1, αIIβ3/Mac-1, ICAM-2/Mac-1, CD40L/CD40, and GPIIb/IIIa/SLC44A2 [75,76] although P-selectin/PSGL1 has been the most prominent interaction found to be involved in direct platelet-induced NETosis [15,18] (Figure 1). Crosstalk between platelets and neutrophils enhance the thrombotic process as neutrophils become activated and consequently undergo NETosis [15], while NETs in turn can promote platelet activation [77,78,79], further perpetuating the thrombotic condition. Circulating neutrophil/platelet complexes have been found associated with inflammation, thrombosis, and sepsis [15,16,80,81]. Platelets can also secrete molecules that activate neutrophils to undergo NETosis such as HMGB1, considered a critical mediator of thrombosis and further promoting platelet aggregation [19,82,83]. 

The glycocalyx of the vascular endothelium provides an anticoagulant separation barrier between blood and tissues. Endothelial inflammation (thromboinflammation) involves the upregulation of adhesion molecules including E-selectin, P-selectin, and intracellular adhesion molecule-1 (ICAM-1) [84], which are critical in the development of thrombotic diseases including stroke and cardiovascular events. Neutrophils are attracted to damaged endothelium, attach via integrins such as Mac-1 [85], and stimulate further inflammation, the formation of neutrophil-platelet aggregates and promote atherosclerosis [86]. NETs stimulate endothelial inflammation through the activity of NETs components such as cathepsin G [87]. In vitro, purified NETs were shown to induce endothelial cell activation [87] in the absence of neutrophils. Franck at al. found that PAD4 deficiency (or DNAse I treatment) prevented endothelial injury and caused a reduction in thrombus formation. This endothelial cell death and detachment was promoted by NETs and mediated by complement activity [88]. Complement function, however, may be dispensable for NETs-mediated damage since ICs can induce vascular inflammation via Fc receptors in the absence of complement [89]. The role of NETs in the endothelium is also supported by observations of inflammation and occlusion in COVID-19 patients associated with NETs formation [90]. Platelets also attach to damaged endothelium and contribute to neutrophil recruitment and further NETs formation and inflammation [84]. 

## 4. Immune Complexes, NETs, and Immune Thrombosis

Thrombosis is the immediate cause of stroke, myocardial infarction, embolism, and deep vein thrombosis. As such, it is one of the principal causes of morbidity and mortality worldwide. More recently, a subtype of thrombus formation in blood vessels resulting from the activity of leukocytes, platelets, and the endothelium has become known as immunothrombosis [91] or thromboinflammation. A common example of immunothrombosis occurs in cases of systemic infection, where generalised inflammation, leukocyte, and platelet activation lead to thrombosis, which contributes to multiple organ failure. NETs contribute to thrombosis by promoting thrombin generation through tissue factor activity, the activation of factor XII and the interaction and activation of platelets via histone activity. NETs also contribute to thrombus stability by providing a DNA scaffold and inhibiting fibrinolysis. The presence of NETs in coronary artery [92,93], stroke [94] and infection-related thrombi [90] provides compelling evidence for the involvement of neutrophils in distinct types of thrombosis (Table 2). 

An early example of IgG ICs generating NETs was provided by Kelley et al. [39]. The authors showed that antineutrophil cytoplasmic antibodies (ANCA) induced NETs via interaction with Fc receptors [39]. Some ANCA antibodies, such as anti PR3 antibodies, may also induce NETs formation independently of Fc interaction [95]. Patients with ANCA antibodies present with vasculitis and have increased incidence of venous thromboembolism [96] and arterial thrombotic events. In fact, NETs formation not only contributes to inflammation in ANCA vasculitis, but also generates autoantigens, such as myeloperoxidase, that contribute to a further development of ANCA antibodies [97]. 

Autoantibodies from APS patients, such as anti β2-glycoprotein I antibodies, induce NETs release mediated by TLR4 signalling [54] and NETs formation was required for thrombosis in a mouse model of APS [98]. Likewise, an enhanced risk of cardiovascular events in lupus erythematosus patients is associated with the presence of complexes such as DNA/anti-dsDNA antibody ICs. Importantly, these ICs induced NETs formation in a Fc receptor-dependent manner and stimulated endothelial cell activation in vitro [71]. Patients with rheumatoid arthritis are at higher risk of deep vein thrombosis and embolism. Some ICs associated with this disease activate FcγRIIA and induce the expression of TNF-α [41]. This is likely associated with endothelial inflammation and induce NETs formation. 

Experimental evidence for the role of ICs in thrombosis has been provided by our studies of HIT and VITT IC-mediated thrombosis [15,16,36]. These studies demonstrated that IgG ICs activate both platelets and neutrophils via FcγRIIA interaction. An important observation from these studies is the indispensable requirement of neutrophils in IC-mediated thrombosis. Ex vivo, thrombosis was inhibited when using neutrophil-depleted blood, despite the presence of platelets. In vivo, ICs were unable to induce thrombosis in the absence of NETosis. Together, the data suggest that IC-induced NETosis is required for thrombosis in HIT and VITT. Unexpectedly, the absence of platelets did not preclude thrombus formation in the HIT mouse model. The interaction of ICs with FcγRIIA on platelets caused thrombocytopenia in HIT and VITT but did not lead to thrombus formation when neutrophil activity was inhibited [15,16]. In this context, other Fc receptors were dispensable since blockage of FcγRIIA was sufficient to prevent NETs formation, thrombocytopenia and thrombosis. 

NETs formation is also common in viral infections and is critical in the pathology of COVID-19 [99,100,101,102]. IgG-spike protein complexes can activate platelets via FcγRIIA [65], which is indicative of IC formation. In fact, IgG ICs have been associated with the inflammatory response and disease severity [103]. The involvement of NETs in COVID-19-related thrombosis is supported by observations of neutrophils and NETs in pulmonary thrombi [63]. The induction of NETs in the context of COVID-19 is mediated, at least in part, by platelet–neutrophil interactions [63].

**Table 2 biology-12-01332-t002:** Various NETs formation-associated stimuli.

Condition	Comments	References
Cardiovascular	AMI patients. Neutrophils are activated in acute coronary syndrome. NETs are formed at culprit lesion site. Stroke patients. Neutrophils and NETs markers detected in retrieved stroke thrombi. NETs are associated with severity and mortality. DVT patients. Elevated levels of activated neutrophils and circulating nucleosomes increase risk of DVT. DVT mouse model. NETs promote thrombus formation.	[19,78,94,104,105,106,107,108,109,110]
Diabetes	Type I diabetes patients and mice. Type II diabetes patients. Platelet/neutrophil aggregates present. Neutrophils are primed to undergo NETosis. Homocysteine elevated in diabetic patients and correlates with NETosis.	[111,112,113,114]
Autoimmunity	APS patients. APS antibodies promote NETs and thrombosis. HIT patients and mouse model. NETs are essential for thrombosis. VITT patients and mouse model. NETs mediate thrombosis. Rheumatoid arthritis patients. NETs mediate cartilage damage. Psoriasis patients. NETs promote inflammation. Lupus erythematosus. Anti-DNA antibodies promote NETosis. Association with endothelial disfunction and cardiovascular disease.	[15,16,47,54,71,115,116,117]
Cancer	Pancreatic adenocarcinoma. NETs contribute to cell migration and invasion. Gastric cancer patients. NETs are involved in metastasis. Breast cancer patients and mouse model. NETs contribute to cancer-associated thrombosis. Chronic myelogenous leukemia mouse model. Neutrophils in CML mice more susceptible to NET formation.	[118,119,120,121,122,123]
Infection	Sepsis patients and mouse model. NETs and NETs components promote coagulation and death. Influenza A patients. Virus-IgA complexes stimulate NETosis and inactivate viruses. NETs induce inflammation. HIV-1 patients. Virus-IgA complexes stimulate NETosis and inactivate viruses. NETs induce inflammation. SARS-CoV2. NETs mediate severe COVID-19 pathology. CHIKV patients. NETs limit viral load. Hantaan virus. Strong NETs stimulation. May contribute to kidney and lung damage. Dengue virus. Viral exosomes induce NETs and promote proinflammatory cytokine release. *S. aureus.* NETs promote intravascular coagulation. *A. fumigatus* *C. albicans*. NETs mediate fungal killing. *E. coli.* NETs promote intravascular coagulation.	[2,21,99,100,101,102,124,125,126,127,128,129,130,131,132,133]

AMI, acute myocardial infarction; DVT, deep vein thrombosis; APS, antiphospholipid syndrome; HIT, heparin-induced thrombocytopenia; VITT, vaccine-induced thrombotic thrombocytopenia; CHIKV, Chikungunya virus.

## 5. NETs Targeting in Immune Thrombosis

Several pathways of NETs formation have already been targeted both pharmacologically and genetically. NETs formation has been prevented by inhibitors of PAD4 (CI-amidine, GSK-484), MPO (PF-1355), elastase (AZD9668), ROS and NOX (DPI), and other compounds (reviewed in [134]). The usefulness of these inhibitors is tempered by certain shortcomings. For instance, PAD4 mice are more susceptible to infection, and the inhibition of ROS production may also have detrimental effects on infections. NETs breakdown with DNase I reduces thrombosis but releases harmful NETs products such as histones and MPO that may cause further tissue damage. Other drugs in clinical use that may inhibit NETosis, albeit more indirectly, include heparin, metformin, clopidogrel, and anti-IL-1β antibody [135]. All these therapies, however, are directed to either NETs themselves, NETs components or signalling pathways. Targeted therapies that would prevent Fc receptor interaction with pathogenic ICs are likely to be more effective and with potentially fewer side effects. Fc receptor blockage has been proposed, and therapies have been developed, for several immune conditions. For instance, intravenous immunoglobulins work, in part, by the overloading and blockage of Fc receptors [136]. Similarly, Fc receptor blockage using Fc fragments has also been used to treat immune thrombocytopenia (ITP) [137]. Even though toxicity was observed following the use of anti-FcγRIIIA blocking antibody in ITP, this was due to the high affinity of the antibody causing receptor cross-linking. This can be overcome by the engineering of antibody fragments with a lower binding affinity [138]. The blockage of FcγRIIA by the monoclonal antibody IV.3 has been shown to completely prevent NETosis and thrombosis mediated by FcγRIIA-activating ICs in vitro and in mouse models of thrombosis [15,16]. This represents a proof-of-concept of the viability of Fc receptor inhibition to prevent immune thrombosis. Recently, an FcRn-blocking antibody fragment (modified Fc fragment) was approved for use for the treatment of myasthenia gravis, an autoimmune neuromuscular disease [139]. This is promising for the future development of Fc blocking agents to prevent IC-mediated cell activation.

## 6. Conclusions

The conventional concept of thrombosis surmises that thrombi are formed due to platelet and coagulation activation. The partial efficacy of anticoagulants and antiplatelets suggest that additional aspects are involved. The pathogenesis of immune thrombosis has evolved significantly over recent decades, with the discovery of NETosis illustrating that mechanisms of thrombosis are multicellular and complex. The growing recognition of NETs involvement in conditions such as infection, cardiovascular disease, cancer, diabetes, and autoimmunity has resulted in an improved understanding of disease mechanisms and the discovery of potential biomarkers and novel therapies. Various NETs inhibitors have been studied in a multitude of conditions, including recently developed oral PAD4 inhibitors. Due to the involvement of autoantibodies and ICs in immune thrombosis, further studies are needed to design therapies that can prevent the interaction of these complexes to receptors on platelets and neutrophils to prevent cell activation and the development of NETosis, thus hindering one of the initial steps of thrombus formation. By continuing to examine the interactions between ICs, Fc receptors, and NETs, more effective treatments are likely to emerge with the aim of reducing thrombotic burden in patients with immune thrombosis. 

## Figures and Tables

**Figure 1 biology-12-01332-f001:**
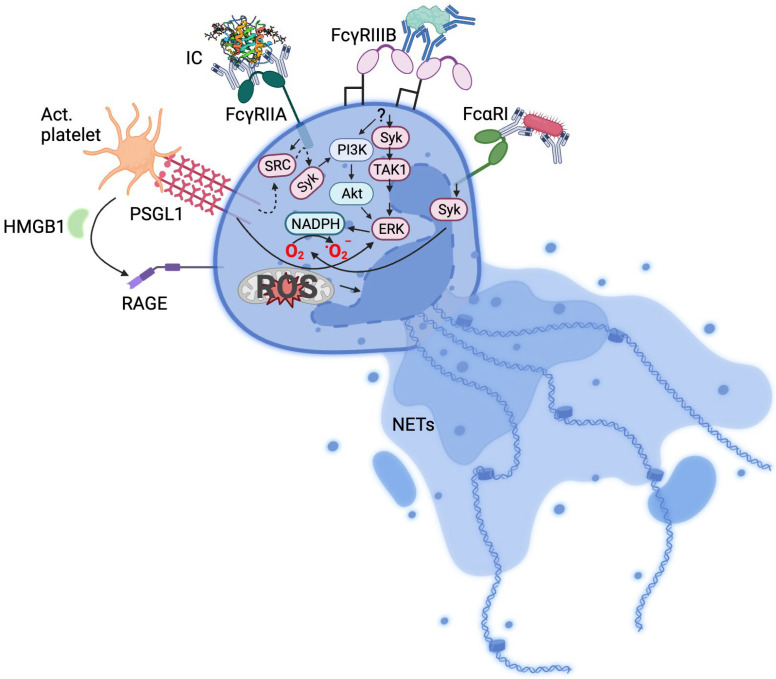
Mechanisms of neutrophil activation. Summarised receptor signalling pathways implicated in NETosis. IC, immune complex; Act. platelet, activated platelets; HMB1, high-mobility group box 1; TAK1, transforming growth factor-β-activated kinase 1; RAGE, receptor for advanced glycation end-products; PSGL-1, P-selectin glycoprotein ligand-1; dotted arrows indicate intermediate signalling molecules not included in the Figure.

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
