# Peer review of "Immune Thrombosis: Exploring the Significance of Immune Complexes and NETosis"

_biology, 2023, doi:10.3390/biology12101332_

Round 1

Reviewer 1 Report

To Author:

The authors summarized the roles and mechanisms of immune complexes and NETs in immune thrombosis in detail in this review paper. I considered this review to be significant. However, I have several suggestions before it can be accepted.

Comments:

(1) I think the Figure 1 should be changed to a table.

(2) In the fourth part, the mechanism of neutrophil activation should be shown with a detailed cartoon diagram to help readers understand more.

(3) I think that the fourth part of the neutrophil activation mechanism should be moved to the third part, and the third part (immune complexes, NETs and immune thrombosis) should be moved to the fourth part, so that the logic of the entire review is more coherent.

Reviewer 2 Report

The authors reviewed the significance of Immune Complexes and NETosis on thrombosis.  The authors provided adequate information on immune complexes, Fc receptors, the mechanism of neutrophil activation, NETosis, and immune thrombosis.

Please see some minor comments and edits that the authors may want to consider.

·      Overall, the mechanism of neutrophil and platelet activation are some things that could have been presented earlier in the paper before discussing “Immune complexes, NETs, and immune thrombosis”

·      Introduction: It would probably be good to provide a little information on platelet-neutrophil interaction especially related to NETosis since platelets have a significant role in thrombosis.

·      Line 126-127: Is it supposed to be a sentence by itself or part of the following paragraph?

·      Table: It would be good to have what type of NETosis is induced in each case by adding another column

·      Figure:

o   The figure as it is not very clear. Are the authors indicating NETosis formation in the tissue that is entering a blood vessel? Are all the diseases listed in the figure contribute to thrombosis due to Ig-mediated NETosis? It could benefit from better labeling. For example, a legend of cells in thrombosis, what type of NETosis is happening, etc..

o   It would also be good to see a visual of Ig-mediated NETosis as a separate figure or as an addition to Figure 1.

Reviewer 3 Report

The review is devoted to the impact of immune complexes (ICs) on NETs-associated thrombosis. Although there are many papers regarding NETs formation, ​​as far as I know this is the first one that deals with NETs in this way (ICs, Fc receptors, NETs, thrombosis) thoroughly, and I think it is important, because ICs are evident/possible in vivo NETs inducer, in many pathological conditions.

The paper is very well-written, informative, and interesting to read.

Suggestion:

Lines 53-56:

“Consequences of NETs formation – anti-bacterial activity, inflammation, vascular occlusion – can, therefore, be attributed to the activity of the diverse components of these DNA structures.”

NETs consist of DNA and proteins, not only DNA; rewrite this sentence.

Round 2

Reviewer 1 Report

The revised paper is ok. I don't have any comments.

Reviewer 2 Report

The authors did a good job of addressing the concerns.